Changes in the spatial and temporal pattern of natural forest cover on Hainan Island from the 1950s to the 2010s: implications for natural forest conservation and management

Lin Siliang
Jiang Yaozhu
He Jiekun
Ma Guangzhi
Xu Yang
Jiang Haisheng jhs@scnu.edu.cn
School of Life Science, South China Normal University , Guangzhou , Guangdong , China
Sandhu Harpinder
Electronic publication date: 2017 May 16
Publication date: 2017
Volume: 5
Electronic Location ID: e3320
Received 2016 Nov 18; Accepted 2017 Apr 15
Copyright: ©2017 Lin et al.
Copyright year: 2017
Copyright holder: Lin et al.
License: This is an open access article distributed under the terms of the Creative Commons Attribution License, which permits unrestricted use, distribution, reproduction and adaptation in any medium and for any purpose provided that it is properly attributed. For attribution, the original author(s), title, publication source (PeerJ) and either DOI or URL of the article must be cited.
License URL: https://creativecommons.org/licenses/by/4.0/

Keywords: Natural forest cover, Dynamic change, Hainan Island, Forest fragmentation, Conservation strategies

Funding: The authors received no funding for this work.

==============================
The study of the past, present, and future state and dynamics of the tropical natural forest cover (NFC) might help to better understand the pattern of deforestation and fragmentation as well as the influence of social and natural processes. The obtained information will support the development of effective conservation policies and strategies. In the present study, we used historical data of the road network, topography, and climatic productivity to reconstruct NFC maps of Hainan Island, China, from the 1950s to the 2010s, using the random forest algorithm. We investigated the spatial and temporal patterns of NFC change from the 1950s to the 2010s and found that it was highly dynamic in both space and time. Our data showed that grid cells with low NFC were more vulnerable to NFC decrease, suggesting that conservation actions regarding natural forests need to focus on regions with low NFC and high ecological value. We also identified the hot spots of NFC change, which provides insights into the dynamic changes of natural forests over time.

Introduction

Human activity is the dominant cause of contemporary environmental change worldwide (Lewis & Maslin, 2015). Land use by humans is a major component of the long-term anthropogenic global changes that have defined the “Anthropocene” as a new epoch of geologic time (Ellis et al., 2013; Dirzo et al., 2014; Lewis & Maslin, 2015). The study of land use and land cover change was initially dominated by monitoring and modelling of the ecological impacts of major land cover changes, such as deforestation and desertification, on the natural system (Lambin & Geist, 2006). However, research of land cover change has now become more integrative, focusing on both the drivers and impacts of land change, including a wider range of interacting processes. Understanding the drivers, state, trends and impacts of different land cover change in combination with the social and natural processes might help to reveal the effects of land system changes on the socio-ecological system and consequently develop effective conservation strategies.

The decrement of natural forest is an important environmental issue in the tropical environment, since these forest harbour exceptional biodiversity, providing important ecosystem services that support the livelihood of local communities (Myers et al., 2000; Sodhi et al., 2006; Page, Rieley & Banks, 2011; Arima, 2016). The study of the past, present, and future state and dynamics of the tropical natural forest cover (NFC) changes might help to better understand the manner of deforestation and fragmentation as well as the influence of social and natural processes. The obtained information will support the development of effective conservation policies and strategies, especially in areas with rapid forest decline (Romijn et al., 2015). The first comprehensive assessment of global forest resources was reported by Zon & Sparhawk (1923), the Food and Agriculture Organization (FAO) of the United Nations has carried out regular assessments (Forest Resources Assessments; FRA) at intervals of five or ten years since 1948 (MacDicken, 2015). Although the FRA data are only partially accurate (Grainger, 2008), they shed light on NFC changes at a global scale (MacDicken, 2015), though not at a local scale, mainly because the mechanisms of NFC changes are more complicated than those reported in previous studies (Lambin, Geist & Lepers, 2003). Remote sensing by satellites has been used to tracking NFC since 1972 (Grainger, 2008). Hansen et al. (2013) developed a dataset of high-resolution global maps and assessed forest cover change from 2000 to 2012 using satellite data with a spatial resolution of 30 m. These results depict a globally consistent and locally relevant record of forest change. However, little is known about the long-term historical distribution of NFC prior to the 1970s (Liu & Tian, 2010). Overall, it is essential to assess the historical distribution of NFC at a specific spatial scale (Miao et al., 2013) in order to explore the underlying mechanisms (Goldewijk, 2001; Goldewijk et al., 2011) and create more effective conservation policies.

Previous studies used biophysical (e.g., topography and climate) and socio-economic data (e.g., population density, road networks, migration, urban extension, and gross domestic product) to reconstruct the long-term historical distribution of NFC (Millington, Perry & Romero-Calcerrada, 2007; Arima et al., 2008; Miao et al., 2013; He, Li & Zhang, 2014). However, these variables are only valid in regions with abundant historical data. Of the variables, roads have long been considered an important driver of deforestation in many tropical countries, because of their high availability and strong correlation with NFC (Arima et al., 2005; Arima et al., 2008; Gaveau et al., 2009; Freitas, Hawbaker & Metzger, 2010; Cai, Wu & Cheng, 2013; Walker et al., 2013; Li et al., 2014; Arima et al., 2016; Hu et al., 2016). Multiple studies have focused on the effect of roads on NFC, but only a few of them have used the temporal and spatial dynamics of road networks for studying the distribution of NFC (Perz et al., 2007; Perz et al., 2008; Ahmed et al., 2013; Newman, McLaren & Wilson, 2014).

Hainan is the largest island in the Indo-Burma biodiversity hot spot and harbours high levels of biological endemism (Myers et al., 2000). Natural forest initially covered almost the entire island, but NFC has now decreased to less than one quarter, mainly in the central mountainous area of the island, owing to the intensive exploitation and deforestation since the 1950s (Lin et al., 2015). Although the NFC changes in Hainan have been previously reported, limited information is available regarding the distribution of NFC over time, which is important to better understand the dynamics of wildlife communities (Brook et al., 2006), especially of some endangered species (e.g., Hainan Gibbon; Xu et al., in press). The deforestation of Hainan Island has led to a series of grave ecological consequences (e.g., large floods, landslides, and drought) that threatened the local communities and decreased the overall environmental quality (Wang et al., 2014). Thus, understanding the spatial and temporal dynamics of NFC is crucial to evaluate the environmental quality and restore forest ecosystems that important services to the local residents.

Here, we used historical road maps and environmental variables to simulate and document the temporal and spatial dynamics of NFC on Hainan from the 1950s to the 2010s. First, we fitted a random forest (RF) model to simulate the distribution of NFC with road-related and environmental variables in 1975, 1995, and 2012. Next, we used the random forest model to reconstruct the NFC at a 20-years intervals from the 1950s to the 2010s. Based on our data, we aimed to: (1) explore the effect of the road network on NFC changes and (2) identify changes in the distribution of NFC from the 1950s to the 2010s.

Material and Methods

Study area

This study was carried out on Hainan Island, an area of approximately 33,900 km2 located in south China (Fig. 1). The island has a central mountainous region with a maximum elevation of 1,867 m (Fig. 1) and is flat in the northern and coastal regions. The climate is tropical with warm and humid weather. The average annual temperature ranges from 16°C in January to 27.5°C in July. The annual rainfall is over 1,600 mm (Francisco-Ortega et al., 2010) and unevenly distributed both between the rainy and dry season and across the island. Rainfall mainly occurs in eastern Hainan Island (Wenchang, Qionghai, Wanning, Lingshui, and Sanya Counties) due to typhoons from the Pacific Ocean. The western part of the island is characterized as dry, while the eastern part is humid (Xu et al., in press).

Figure 1 Location and topography of Hainan Island, China.

JFL, Jianfengling Mountain; BWL, Bawangling Mountain; YGL, Yinggeling Mountain; WZS, Wuzhishan Mountain; DLS, Diaoluoshan Mountain.

Prior to 1950s, the forest area on Hainan Island was largely natural, since the number of coconut and rubber plantations was limited (Chen, 1948), whereas the non-forest area was mainly farmland. The natural forest area decreased considerably owing to logging for timber, rubber plantations, and residential expansion, with NFC reaching a minimum in the 1980s. In response to this dramatic NFC loss, the Hainan Provincial Government established a natural forest logging ban in 1994 (Standing Committee of Hainan Provincial People’s Congress, 1993; Zhang, Uusivuori & Kuuluvainen, 2000).

Distribution of natural forest

In this study, arboreal forest, shrub land, and open woodland were considered as natural forest based on the standards of the National Forest Inventory of China (SFA PRC, 2011). The distribution of natural forest in 1975 was obtained from 93 topographical maps of Hainan Island and digitised using ArcInfo 9.3 (ESRI, 2008; see Appendix S1 for more details). These maps incorporated different types of land uses and outlined the distribution of arboreal forest, shrub land, and open woodland. The distribution of natural forest in 1995 and 2012 was reproduced from maps of the 5th and the 8th National Forest Resources Inventory of China (http://www.forestry.gov.cn/gjslzyqc.html). Three datasets of the historical natural forest distribution were primarily produced by aerial photography and validated by field observations. Field investigation data of 1997/1998 and 2012/2013 (Xu et al., in press) were used to estimate the accuracy of natural forest distribution in 1995 and 2012 (Fig. S1). To enhance data consistency, the NFC distribution in the 1970s, 1990s and 2010s was combined using the ‘Simplify polygon’ function in ArcInfo 9.3 (ESRI, 2008).

We established a 5 × 5 km grid cell system for model simulations. First, Hainan Island was divided into 5 × 5 km grid cells, and those with a land area less than 50% were excluded to minimise the bias of low NFC. A total of 1,360 grid cells were used for further analysis. Next, we overlaid the polygons of natural forest in 1975, 1995 and 2012 with the grid cells, and the proportion of natural forest was considered as the NFC of grid cell. The obtained NFC datasets were used as a response variable in model simulations.

Variables correlated with NFC

We investigated three main categories of variables with a potentially high correlation with NFC: (1) topography, such as slope (SLO, °) and elevation (ELE, m), which indicate difficulty in accessibility and historical exploitation (Teixeira et al., 2009); (2) climatic potential productivity (CPP, Kg  hm−2 a−1), which indicates the potential of forest loss due to crop farming (Pongratz et al., 2008); and (3) road network, which indicates the scale and intensity of socio-economic impact on NFC (Arima et al., 2005; Walker et al., 2013; Van der Ree, Smith & Grilo, 2015; Arima et al., 2016).

Digital elevation data (approximately 90 × 90 m resolution) were obtained from the Consortium for Spatial Information (http://www.cgiar-csi.org/data/srtm-90m-digital-elevation-database-v4-1). The slope map was created based on changes in elevation between adjacent pixels of the digital elevation data. Data on the climatic potential productivity (approximately 1 × 1 km resolution) were obtained from the National Earth System Science Data Sharing Infrastructure (http://www.geodata.cn/index.html). The SLO, ELE, and CPP of each grid cell were calculated by averaging all cells (1 × 1 km or 90 × 90 m) such that their centroid fell within each 5 × 5 km grid cell.

Historical data of the road network were obtained from maps and digital databases published prior to 2013 (Table S2). Maps were digitised using ArcInfo 9.3. (ESRI, 2008) and assigned to a 20-year interval. Due to differences in the mapping scale of traffic maps, the Rural road was designated as the lowest level road, whereas Expressway, Highway, Simply-built highway, and Cart road were designated as roads with relatively higher levels (Table S2). Based on the function and surface material of various roads, we classified these roads into two classes in the 1950s, 1970s, 1990s and 2010s: main road (Expressway, Highway, and Simply-built highway) and secondary road (Cart road and Rural road).

Four road-related variables, including the sum of road length (SRL, km km−2), the distance from the grid cell centroid to the nearest road (DNR, km), the number of nodes of road network (NON, ea km−2), and the mean node degree (MND, ea km−2; the mean number of roads connected to each node), were calculated in each grid (Table S3) to represent the density and configuration of the road network in the 1950s, 1970s, 1990s, and 2010s.

Reconstruction of the historical NFC from the 1950s to the 2010s

Prior to the reconstruction of the historical NFC, we tested the predictive performance of four modelling techniques, including the generalized linear model (GLM), generalized additive model (GAM), artificial neural networks (ANN) model, and random forests (RF) model. We used the normalised mean square error (NMSE) to compare the predictive performance of the four models in simulating the NFC of each 5 × 5 km grid cell (Appendix S5). NMSE is a relative measure, estimating the overall deviation between predicted and measured values and also a unit-less measure in the interval [0, 1]. The modelling technique with the lowest value of NMSE has the highest predictive performance. The RF model showed the lowest NMSE among the four models and consequently, was used for reconstructing the historical NFC from the 1950s to the 2010s (Table S4).

The RF model (Iverson et al., 2008) is characterized by insensitivity to multicollinearity, steadiness across unbalanced datasets, accuracy in predicting the effects of multiple explanatory variables (Breiman, 2001; Culter et al., 2007), higher efficiency in large datasets than that of traditional machine-learning techniques, suitability in demonstrating the nonlinear effect of variables, modelling complex interactions among variables, and robustness to outliers (Li, 2013; Breiman, 2001; Culter et al., 2007). In the present study, we fitted the RF model to associate the NFC data of each grid cell with SLO, ELE, CPP, and main and secondary road networks related indicators (SRL, DNR, NON, and MND) in the 1970s, 1990s, and 2010s. Next, we used the obtained RF model to reconstruct the historical NFC of each grid cell at 20-year intervals from the 1950s to the 2010s, according to the topography, climate, and road network data in a specific period. We assumed that topography and climate did not change significantly over a relatively short time and therefore used the same SLO, ELE, and CPP in all the assessed time periods (1950s–2010s, 1950s–1970s, 1970s–1990s, and 1990s–2010s). The methodological sequence describing the input data, the modelling and validation approaches, and the generation of predictions is shown in the Fig. 2.

Figure 2 Flowchart showing the methodological sequence describing the input data, the modelling and validation approaches, and the generation of predictions.

Overall, we investigated the historical, temporal, and spatial dynamics of NFC on Hainan Island from the 1950s to the 2010s. We first applied simple linear regression to explore the relationship between each road-related indicator and the NFC change rate of each grid cell in the four assessed time periods. We used adjusted R2 to estimate the explanatory power of each road-related indicator in the simple linear regression model. The variation partition was also applied to compare the effects of main roads vs. secondary roads on NFC change. Simple linear regressions with the combined effect of the four road-related indicators of main and secondary roads were used to estimate the variation partition of different road types in the four assessed time periods.

All analyses were conducted in R 3.2.3 (R Development Core Team, 2016) using the ‘mgcv’ (Wood, 2011), ‘caret’ (Kuhn, 2016), ‘randomForest’ (Liaw & Wiener, 2002), ‘DMwR’ (Torgo, 2011) packages.

Hot spots analysis

To quantify the degree of deforestation, the Getis-Ord Gi* statistic was computed to measure the degree of spatial clustering of a local sample and its difference from the expected value using the sum of the differences between values in the local sample and the mean, and is standardized as a Z score (Scott & Warmerdam, 2005). The Z score reveals the spatial clustering of features with either high or low values. To test the significance of the Z score the value within a specific confidence level were compared. Positive Gi* values represent clusters that are greater than the mean (Reforestation hot spots), whereas negative Gi* values represent clusters that are lower than the mean (Deforestation Cold-spots; Getis & Ord, 1996; Cohen et al., 2011). This statistic represents the frequency of a hot spot relative to the number of input deforested points covered the area. To identify spatial clustering patterns of NFC changes at 20-year intervals, hot spots analysis was performed using Getis-Ord Gi* statistics (Mitchell, 2005) with the ‘Hot Spot Analysis’ function in ArcInfo 9.3. (ESRI, 2008).

The Getis-Ord Gi* was calculated as follows:

Gi∗=∑j=1nwi,jxj−X¯∑j=1nwi,jn∑j=1nwi,j2−∑j=1nwi,j2n−1S

where xj is the attribute value for feature j, wi,j is the spatial weight between feature i and j, and n is equal to the total number of features. The Gi∗ statistics is a Z score and thus, no further calculations are required.

Results

The temporal and spatial dynamics of NFC

NFC changes at 20-year intervals from the 1950s to the 2010s are shown in Fig. 3. We also calculated the total NFC to explore the temporal dynamics of NFC from the 1950s to the 2010s. The trend of total NFC indicated a considerable decrease from 41.4% in the 1950s to 24.2% in the 2010s (Fig. 4; R2 = 0.99, p = 0.004).

In the 1950 s, most grid cells with NFC >50% were identified in the central mountainous regions, whereas those with NFC <20% were identified in the coastal regions (Fig. 3). The grid cells with NFC ⩽20% showed a significant increase from 433 to 839 from the 1950s to the 2010s (Fig. 3), whereas their distribution gradually shifed from the coastal regions to the mountainous regions (Fig. 3). The grid cells with NFC >50% mainly distributed in the central and southern mountainous region, but showed a significant decrease from 544 to 286 from the 1950s to the 2010s (Fig. 3). These results revealed that the natural forest decrease and also fragmentized from the 1950s to the 2010s.

Figure 3 Distribution patterns and histogram statistics of natural forest cover (NFC) in the 1950s (A), the 1970s (B), the 1990s (C) and the 2010s (D) on Hainan Island.

Figure 4 Trend of total natural forest cover (NFC) from the 1950s to the 2010s on Hainan Island.

We used the slope of site-specific relationships between NFC and its change rate over time to identify the dynamic change patterns of NFC. The slopes were positively correlated from the 1950s to the 2010s, indicating that a lower NFC tended to correspond to a higher rate of NFC decrease (Fig. 5; F = 122.3, p < 2.2 × 10−16). Assessing the pattern of NFC change rate revealed that most grid cells with low NFC change rate were distributed in the mountainous areas (Jiangfengling, Yinggeling, and Wuzhishan mountains), whereas most grid cells with high NFC change rate were distributed in the Qiongbei Platform (Fig. 5). Hot spots analysis from the 1950s to the 2010s showed that NFC decrease mainly occurred in the platform area, whereas NFC increase occurred in the mountainous region (Fig. 6). However, NFC also increased in Wenchang County, although this area is flat and well developed. When assessed at 20-year intervals, the distribution of hot spots with decreased NFC was observed to be shifted from the eastern regions to the western regions of Hainan Island (Fig. 6).

Figure 5 The site-specific relationships between NFC and its change rate (A), and the distribution pattern of change rate (B) from the 1950s to the 2010s.

Green represents high NFC (≥60%) in the 1950s and had NFC increased from the 1950s to the 2010s; Red represents low NFC (≤20%) in the 1950s and had NFC decreased from the 1950s to the 2010s.

Figure 6 Hot spots of natural forest cover (NFC) decrease and increase in four assessed periods: 1950s–2010s (A), 1950s–1970s (B), 1970s–1990s (C), and 1990s–2010s (D) on Hainan Island.

Dynamics of road expansion

The density and configuration of the main road network markedly changed with the time from the 1950s to the 2010s. The mean SRL of main roads increased from 0.11 km km−2 in the 1950s to 0.45 km km−2 in the 2010s (Table 1). The standard deviation of SRL of main roads also increased, indicating that the distribution of main roads became more heterogeneous across the grid cells. The mean DNR decreased from 0.30 km in the 1950s to 0.04 km in the 2010s (Table 1). The standard deviation of DNR also decreased, indicating that the spatial distribution of main roads markedly increased. The average value of NON and MND of main roads increased from the 1950s to the 2010s (Table 1), revealing that the configuration of the road network became more complicated across the grid cells. The standard deviation of NON of main roads increased from 0.04 ea km−2 to 0.23 ea km−2 from the 1950s to the 2010s, whereas the standard deviation of MND of main roads had slightly decreased from 0.05 ea km−2 to 0.04 ea km−2, revealing that the configuration of the main road network became more complex and the spatial pattern of main road was uneven across the grid cells.

Table 1 Changes in road-related variables of main and secondary roads from the 1950s to the 2010s on Hainan Island.

Data are the means and standard deviation of all grid cells.

		1950s	1970s	1990s	2010s	
SRL	Main road	0.11 ± 0.17	0.23 ± 0.22	0.27 ± 0.24	0.45 ± 0.27	
Secondary road	0.18 ± 0.18	0.36 ± 0.23	0.36 ± 0.23	0.36 ± 0.23	
DNR	Main road	0.30 ± 0.29	0.09 ± 0.08	0.08 ± 0.08	0.04 ± 0.04	
Secondary road	0.11 ± 0.11	0.05 ± 0.05	0.05 ± 0.04	0.05 ± 0.04	
NON	Main road	0.01 ± 0.04	0.09 ± 0.20	0.11 ± 0.20	0.20 ± 0.23	
Secondary road	0.03 ± 0.06	0.23 ± 0.21	0.23 ± 0.21	0.23 ± 0.21	
MND	Main road	0.02 ± 0.05	0.04 ± 0.04	0.05 ± 0.04	0.07 ± 0.04	
Secondary road	0.03 ± 0.05	0.05 ± 0.03	0.05 ± 0.03	0.05 ± 0.03	
Notes.

SRL sum of road length (km km−2)

DNR distance from the grid cell centroid to the nearest road (km)

NON number of nodes of road network (ea km−2)

MND mean number of roads connected by each node (ea km−2)

The SRL of secondary roads increased from 0.18 km km−2 in the 1950s to 0.36 km km−2 in the 1970s, but remained stable after the 1970s. The NON and MND of secondary roads also showed the same trend both clearly increase in the 1950s to 1970s, but remained stable after the 1970s (Table 1). The DNR of secondary roads had decreased from 0.11 km in the 1950s to 0.05 km in the 1970s, whereas only slightly changed after the 1970s. The standard deviation of the four road-related indicators showed that the secondary road network markedly changed in for the 1950s to 1970s, but remained stable after the 1970s (Table 1). The density and configuration of the secondary road network markedly changed from the 1950s to 1970s, whereas it changed only slightly after the 1970s (Table 1).

Both the main road and secondary road network obviously developed from the 1950s to the 2010s; however, the construction processes were faster in the former than the latter. Overall, the density and configuration of the road network was higher and more complex after nearly 60 years of construction.

Effects of different road expansion on NFC change

The change of SRL of main roads explained 11.20% (p < 0.001) of the variance in NFC change from the 1950s to the 2010s and was also the strongest indicator in the other three assessed time periods (1950s–1970s, Adjusted R2 = 14.22%, p < 0.001; 1970s–1990s, Adjusted R2 = 20.73%, p < 0.001; and 1990s–2010s, Adjusted R2 = 4.89%, p < 0.001; Table 2). In the 1950s–2010s, the overall variation partition of main roads (Adjusted R2 = 23.40%; p < 0.001) in the simple linear regression was much higher than the secondary roads (Adjusted R2 = 13.30%; p < 0.001). The change of main roads also explained a higher degree of the variance than the change of secondary roads in the other three assessed time periods. These results indicated that the main road network had a more significant effect on deforestation than the secondary road network from the 1950s to the 2010s.

Table 2 The explanatory power (Adjusted R2, %) of the change of road-related indicators in a simple linear regression model.

Change of road-related indicators	Change of NFC in the 1950s–2010s	Change of NFC in the 1950s–1970s	Change of NFC in the 1970s–1990s	Change of NFC in the 1990s–2010s	
Type of road	Indicators	Adjusted R2	p value	Adjusted R2	p value	Adjusted R2	p value	Adjusted R2	p value	
Main road	Change of SRL	11.20	<0.001***	14.22	<0.001***	20.73	<0.001***	4.89	<0.001***	
Change of DNR	11.07	<0.001***	0.27	0.031*	5.97	<0.001***	0.45	0.007***	
Change of NON	4.83	<0.001***	4.37	<0.001***	19.03	<0.001***	1.76	<0.001***	
Change of MND	5.59	<0.001***	8.31	<0.001***	7.35	<0.001***	2.30	<0.001***	
Overall	23.40	<0.001***	17.67	<0.001***	25.20	<0.001***	5.10	<0.001***	
Secondary road	Change of SRL	0.32	0.021*	0.14	0.089	0.36	0.015*	<0.01	0.613	
Change of DNR	7.14	<0.001***	5.26	<0.001***	<0.01	0.801	0.32	0.021*	
Change of NON	5.11	<0.001***	2.26	<0.001***	<0.01	0.932	0.01	0.422	
Change of MND	0.22	0.045*	1.11	<0.001***	<0.01	0.505	0.01	0.605	
Overall	13.3	<0.001***	8.99	<0.001***	0.31	0.085	0.12	0.228	
Main road & Secondary road	30.7	<0.001***	22.5	<0.001***	26.0	<0.001***	5.4	<0.001***	
Notes.

SRL sum of road length

DNR distance from the grid cell centroid to the nearest road

NON number of nodes of road network

MND mean number of roads connected by each node

The p value represents the significance level of the relationship between NFC change rate and change of road-related indicators.

* p < 0.05.

** p < 0.01.

*** p < 0.001.

In the 1990s–2010s, the adjusted R2 of the main road network decreased from 25.2% to 5.10% (Table 2), whereas that of the secondary road network decreased from 8.99% to 0.12% (Table 2), showing that the association between the road network and deforestation weakened with time. However, the main road network still showed stronger association with NFC change than the secondary road network (5.10% vs. 0.12%).

Discussion

NFC reduction and fragmentation

Our results demonstrated that the total NFC of Hainan Island decreased significantly from the 1950s to the 2010s (Fig. 4), and the distribution of grid cells with NFC <20% increased in the mountainous regions and thus became more vulnerable to deforestation (Fig. 5). These patterns are consistent with those reported in other tropical regions, in which natural forest deforestation is always accompanied by fragmentation (Pimm, 1998; Laurance et al., 1998; Defries et al., 2005). Natural forest that is severely fragmented shows small patches of natural forest that are highly vulnerable to clearing (Stickler et al., 2013; Taylor, 2013). We also observed that the grid cells with NFC >50% were located in areas with a high altitude and steep slope (Fig. 3), in which mechanisation and accessibility are limited (Freitas, Hawbaker & Metzger, 2010; Hu et al., 2016). Most nature reserves on Hainan Island were similarly distributed in the mountainous regions with high altitude and steep slope (Lin et al., 2015). Although some grid cells located in some mountainous regions showed a moderately high degree of recovery (Fig. 5), most of them with NFC >50% was shrank and fragmented.

Natural forest fragmentation is a major issue in tropical environments, since the spatial arrangement and geometric configuration of fragments can impair ecological processes. Six general fragmented patterns have been recognised at the global scale in previous work (Geist & Lambin, 2002; Mertens & Lambin, 1997), of which five are observed in the Amazon tropical forest–rectangular, fishbone, radial, dendritic, and ‘the stem of the rose’ (Arima et al., 2005). Road network architecture plays a critical role in shaping forest fragmentation patterns in the Amazon (Arima et al., 2005; Soares-Filho et al., 2006; Walker et al., 2013). We also revealed that the road network expansion explained more than 30% of natural forest deforestation and fragmentation on Hainan Island from the 1950s to the 2010s (Table 2; Fig. 5). When the road network was overlaid by grid cells with high NFC (Appendix S6; Fig. S2), we found that the dominant fragmented pattern of Hainan Island natural forest was radial and dendritic. Unlike the Amazon, road construction combined the complex topography determined the pattern of natural forest landscape on Hainan Island (Fig. 1). Thus, forest fragmentation patterns on the Hainan Island were highly affected by road construction and topography.

Our analysis showed that hot spots with decreased NFC mainly occurred in the Qiongbei Platform (Fig. 5), following a dynamic spatial distribution pattern from east to west, whereas hot spots with increase NFC occurred in Wenchang County (Fig. 5), an area that is flat and well developed with a frequent typhoon occurrence during the summer. The local residents preserved some small patches of natural forest around the villages and buildings, known as “geomantic forest”, as a protecting shield against the severe weather events.

Effect of road network on natural forest deforestation

Road-related indicators are not the actual drivers of NFC decrease, but they play a role in natural forest dynamics through land-use changes and deforestation (Freitas, Hawbaker & Metzger, 2010; Newman, McLaren & Wilson, 2014). The development of roads and road networks is strongly correlated with the economic growth and is associated with the ecological disturbance and natural forest degradation (Wilkie et al., 2000; Laurance, Goosem & Laurance, 2009). In tropical forests, roads have a low direct effect on habitat loss, but a high indirect effect on the spatial patterns of deforestation (Fearnside, 2008), since they facilitate accessibility, resource extraction, and human activities (Selva et al., 2011; Hu et al., 2016). Our results were consistent with those of previous studies that also reported the significant effect of road networks on natural forest deforestation in tropical regions (Perz et al., 2007; Gaveau et al., 2009; Freitas, Hawbaker & Metzger, 2010; Cai, Wu & Cheng, 2013; Li et al., 2014; Arima et al., 2016; Hu et al., 2016). We demonstrated that road-related indicators played a substantial role in determining the deforestation of NFC in Hainan. The SRL of main roads was one of the strongest variables associated with NFC change dynamic in the four assessed time periods (Table 2). However, the power of road network in explaining the NFC reduction weakened over time, in both main and secondary roads, probably revealing the underlying socio-economic processes of road expansion itself (Perz et al., 2007).

The influence of road networks on deforestation is associated to population migration and various socio-economic events. Large populations moved to Hainan Island, since it is area rich in natural resources (Yan, 2008). After the foundation of People’s Republic of China in 1949, the economy became a key element in the national strategic plan. Both agriculture and forest farming developed rapidly, and large agricultural populations migrated to Hainan Island under the conduct of the national government. In the 1960s and 1970s, the ‘sent-down youth’ movement influenced the social activity and increased migration to Hainan. In the 1950s–1970s, roads were constructed for accessing land and timber resources and thus, the main road and secondary road network was markedly associated to natural forest deforestation (Table 2). In 1988, Hainan Island was established as a provincial agency and Economic Development Zone, leading to another population migration event, mainly of urban population. In the 1970s–1990s, roads were built for accessing land resources and linking major cities and towns; thus, the main road network had a stronger influence on deforestation than the secondary road network (Table 2). In the early 1990s, the industrial structure of Hainan Island was transformed and upgraded. The tertiary industry, especially tourism, became the core of economic development, changing the functions of road construction. Additionally, the Hainan Provincial Government established a natural forest logging ban in 1994 in response to the sharp decline in NFC (Standing Committee of Hainan Provincial People’s Congress, 1993). Consequently, the association between the road network and deforestation weakened in 1990s–2010s (Table 2).

Deforestation was more affected by the main road network than the secondary road network in the four assessed time periods (Table 2). Population was the most important factor affecting NFC on Hainan (Zhang, Uusivuori & Kuuluvainen, 2000; Lin & Zhang, 2001). The main road network of Hainan Island includes expressways, highways, and simply-built highways built or funded by the national or provincial governments for geopolitical purposes, especially for connecting major cities. The secondary road network includes cart roads and rural roads built or funded by the city and county governments for supporting local livelihoods, connecting communities, and accessing land and other natural resources. The construction of main roads improves the connectivity of major cities, supports urbanisation (Perz et al., 2008), and thus, highly increases deforestation. The construction of secondary roads also promotes deforestation, but in a smaller scale than that of main roads (Fig. S2).

Implications for natural forest conservation and management

Our study provided significant information for natural forest conservation and management. At a global scale, ecosystems decline and become fragmented (Saunders, Hobbs & Margules, 1991; Fischer & Lindenmayer, 2007), and thus the small natural forest patches represent large elements of tropical natural forest (Tulloch et al., 2016). Theoretical and experimental studies have highlighted the importance of conserving large contiguous natural forest patches for maintaining biodiversity (Bender, Contreras & Fahrig, 1998; Mortelliti et al., 2014). Consequently, conservation actions have mainly focused on preserving large contiguous forest patches and not relatively small fragmented forest patches (Ovaskainen, 2002; Tulloch et al., 2016). Small fragmented forest patches contribute to short-term and long-term indigenous species persistence; enhance the biodiversity in human-dominated fragmented landscapes (Turner & Corlett, 1996); positively affect the diversity of forest bird and mammal species, forest plant species, and migratory animals as well as the persistence of meta-populations (Laurance, 1994; Warkentin, Greenberg & Salgado Ortiz, 1995; Jacquemyn, Butaye & Hermy, 2001); act as refuges for plant and animal species; and help conservationists to launch a final attempt to rescue endangered species that may serve as sources of natural forest reconstruction (Turner & Corlett, 1996). In Hainan, small natural forest patches are formed from the fragmentation of larger contiguous natural forest patches. Those small natural forest patches are isolated and always have an area less than 10 ha in Hainan. Those small natural forest patches are always located at the edge of large contiguous natural forest in the mountainous region or around the villages in the flat region, and provide important ecosystem services for local residents. For example, the “geomantic forest” in Wenchang and Qionghai counties is an example of small natural forest patches with typical tropical forest structure, but high plant diversity (Yang & Wu, 2002), distributed in the Lingnan region that includes the modern Chinese provinces of Jiangxi, Hunan, Guangdong, Guangxi, and Hainan. The geomantic forest has been preserved according to feng shui tradition, which includes historical and cultural features of ecological significance, and provides important ecosystem services such as water and soil conservation as well as protection against severe weather events (Cheng, He & Liu, 2009). Due to this forest, the NFC of Wenchang county increased from the 1950s to the 2010s (Fig. 5). Thus, the protection of small natural forest patches is important, and long-term data need to be used for developing effective management with specific conservation objectives. Regions with low NFC tend to be more vulnerable in natural forest loss (Fig. 5). The natural forest loss and fragmentation patterns are dynamic on Hainan (Fig. 6). Our results showed that the spatial and temporal change patterns of NFC could be accurately assessed using historical data (Newman, McLaren & Wilson, 2014). Thus, conservationists and decision-makers need to evaluate natural forest changes and develop effective management actions (Margules & Pressey, 2000). On Hainan Island, regions with low NFC, but high ecological value (Yangshan region in Haikou city; the Geomantic forest in Wenchang city; Fig. 3) located in flat areas that are well developed and have high population density, are more vulnerable than natural forest within the nature reserves. Thus, conservation frameworks and forest restoration projects are needed to preserve or recover the natural forest.

The NFC in the mountainous region increased in the past 60 years; however, regions with high NFC are vulnerable to fragmentation due to the expansion of the road network in some remote regions in the mountains (Fig. S2). The south-central mountainous region of Hainan Island is a biodiversity hot spot (Myers et al., 2000) and thus, a priority area for conservation in China (Ministry of Environmental Protection of the People’s Republic of China, 2013). Monitoring the natural forest in using remote sensing and drones could help in protecting the natural forest and predict any future changes (Paneque-Gálvez et al., 2014).

Conclusions

This study used historical data to reconstruct NFC maps in order to better understand the spatial and temporal change patterns. The road network was identified as an important factor of forest loss and fragmentation of tropical forests. Although the exact distribution of natural forest was not depicted, we found that the dominant fragmented pattern of Hainan Island natural forest is radial and dendritic.

Our study showed that: (1) low NFC corresponds to a high rate of NFC decrease as well as highly dynamic spatial and temporal change patterns of NFC; (2) the road network significantly affected NFC, whereas topography affected deforestation by influencing the road network construction pattern; and (3) the effect of road network on NFC was related to population migration and socio-economic events, and the main road network had a stronger relation to deforestation than the secondary road network.

Deforestation and fragmentation still occur in Hainan, and public concern remains pronounced in recent years. Thus, it is crucial to better understand implications of deforestation on biodiversity conservation, as well as the driving forces of deforestation. Conservation efforts must focus on small natural forest patches using new technologies (e.g., remote sensing and drones) that will improve monitoring and data collection.

Although our model was incomplete, since it only included topography, climate and road network data, it revealed the spatial and temporal change patterns of NFC on Hainan Island. Future modelling studies need to build an improved model, combining historical, natural, and socio-economic factors that will help to better understand the underlying mechanisms of NFC change patterns.

Supplemental Information

Appendices Supplementary Materials

Appendix 1 Description of obtaining the NFC in 1975 from 93 topographical maps

Appendix 2 Description of the field investigation for estimating the accuracy of natural forest distribution

Appendix 3 The sources of historical road network maps and road levels

Appendix 4 Description of the measure of four road-related indicators

Appendix 5 Comparing the predictive performances of four model techniques

Appendix 6 Overlaying the distribution of NFC with the road network in four pattern predictive performances of four model techniques

Click here for additional data file.

We thank the Forestry Department of Hainan Province, the Administration Bureau of Animal and Plant Protection of Hainan Province, and the staff of Nature Reserves for the provided data and technical support. We are also grateful to the anonymous referees for their valuable comments that helped to improve our manuscript.

Additional Information and Declarations

Competing Interests

Author Contributions

Data Availability

The authors declare there are no competing interests.

Siliang Lin performed the experiments, analyzed the data, wrote the paper, prepared figures and/or tables.

Yaozhu Jiang performed the experiments, analyzed the data, contributed reagents/materials/analysis tools, wrote the paper, prepared figures and/or tables.

Jiekun He wrote the paper, prepared figures and/or tables.

Guangzhi Ma and Haisheng Jiang conceived and designed the experiments, reviewed drafts of the paper.

Yang Xu contributed reagents/materials/analysis tools.

The following information was supplied regarding data availability:

The raw data has been supplied as a Supplementary File.

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
