# Peer review of "Changes in the spatial and temporal pattern of natural forest cover on Hainan Island from the 1950s to the 2010s: implications for natural forest conservation and management"

_PeerJ, doi:10.7717/peerj.3320_

## Round 0.1 · original submission · Major Revisions

· Academic Editor

Major Revisions

I have received three reviews of your manuscript. While one reviewer suggests minor changes to the paper, two reviewers who are more familiar with the methods and location suggests major revisions. In my opinion these changes can be incorporated by paying more attention to the comments by all three reviewers especially about explaining methods in details.

Reviewer 1 ·

Basic reporting

No Comments

Experimental design

Most importantly, the methodology is not explained in any detail. Thereby making it impossible to review, analyse, and if needed duplicate the methods involved. For example, in lines 93-94 how the distribution of NFC in 1975 obtained from 93 topographical maps? how many levels of road included in the road network database and whether or not different NFC maps would be by using the different level roads in the random forest algorithm.In particular, the authors need to specify the random forest model modelling verification in this study.

Validity of the findings

The findings of spatial and temporal patterns of NFC change from 1950 to 2010 are interesting/ However, because of the short in the methods description, the results are not appropriately sounded. The implications for natural forest conservation and management are also need to give more careful consideration. for example, which is the definition of “small natural forest patches”? which is the equilibrium relationship between the local livelihood and the natural forest protection, is it necessary to protect all the “small” natural forest patches? more elaborate suggestion should be given in the conservation actions regarding natural forests.

Additional comments

The paper " Changes in the spatial and temporal pattern of natural forest cover in Hainan Island from 1950 to 2010:implication for natural forest conservation and management" presents information on the long-term changes in natural forest cover in Hainan. It has a number of interesting elements including tropical forests cover, historical data of the road network, and random forest algorithm etc. Also, the field investigation database in 1997-1998 and 2012-2013 are valuable. At the same time, the manuscript suffers from major scientific shortcomings.

Reviewer 2 ·

Basic reporting

This article meets the basic reporting criteria. The self contained criteria is questionable as assumptions are made that readers know the'random forest model' assumptions and data requirement.

Experimental design

There are main points that need to be expanded on 1) cell sizes (resolution issues), 2) random forest model, and occasional run-on sentences

1) What is (5 × 5 km or 90 × 90 m) mean as stated on line 124? I think a flowchart of your process would help the reader and clarify where the different resolutions are being used. There is a lot of literature about spatial resolution and it seems as though it is of no concern in this research. I think the paper needs to be clearer and more transparent about use of different resolutions.

2) Just giving a citation of a model is not enough in a paper. You need to explain to the reader who may or may not have access to the original paper what the model is and why it is valid for your research. Not only what the limitation are.

3) See line 219 for an example.

Validity of the findings

The discussion is fine. However, there is a missing conclusion or summary of findings. There needs to be a comprehensive synthesis of the research at the end of the paper. I might suggest even section on future research or use for the research in other environments.

Additional comments

One of my comments often is that the authors provide a flowchart for the methods especially when working with multiple datasets. There are potential temporal and spatial inconsistency in the paper and a flowchart would help clarify what time and resolution is being used at each of the step of analysis.

·

Basic reporting

The article is correctly structured and self contained. While the understanding is clear the English does need a final read thru by a native English speaker before the revised manuscript can be accepted.


Hainan Island should make for an intriguing case study on deforestation over time, and the promise of a strong element on native forest elements of the landscape on Hainan Island would make this paper interesting to ecologists at least, and the land use change community. However, that promise does not, make this paper novel or innovative by any means, but documenting an interesting case study might be acceptable for publication.

The changes in NFC outlined on the discussion are intuitive and while this research has measured these there is hardly anything new for readers outside Hainan Island from a general viewpoint. A similar argument can be had regarding the rates of NFC in the next paragraph. This line of arguments can be extended to the discussion on fragmentation and the influence of the road network in loss of NFC.

However, the article disappoints on many fronts. It is not acceptable in its current format and requires minor amendments, which are listed with line references in the accompanying pdf), and significant expansion to arguments as follows.

The context of this paper needs to be widened extensively so that is expanded from a rather pedestrian case study with no methodological innovation or insights that researchers who are interested in these issues outside Hainan Island can learn from. That is the standard of a paper in an international journal. I strongly encourage the authors to read more widely, and set their study in wider contexts – there are three contexts – the objectives of the Global Land Program, global conservation concerns, and/or ecosystem services in humid tropical landscapes. Without at least one of these being invoked and used throughout the paper the article would not, in my opinion be acceptable for publication.

Experimental design

Original primary research. The research question needs to improved, so material in quotation marks in 1 above - no knowledge gap is identified. There are technical issues that I have identified in the attached file, the most important of which is considering the different types of roads in the road network used in the analysis.

Validity of the findings

Again, see my comments in quotation marks in 1 above. Impact is very limited and novelty is lacking entirely. Only very marginal benefit to existing literature. Some questions about robustness of data (see line by line comments in pdf). Discussion and conclusions are pedestrian.

---

## Round 0.2 · accepted · Accept

· Academic Editor

Accept

I am satisfied with the revisions made as suggested by the reviewers and recommend this paper for publication. Please correct some spelling errors in figures/tables during the production stage.